# Symmetry-Like Relation of Relative Entropy Measure of Quantum Coherence

**DOI:** 10.3390/e22030297

**Published:** 2020-03-05

**Authors:** Chengyang Zhang, Zhihua Guo, Huaixin Cao

**Affiliations:** School of Mathematics and Information Science, Shaanxi Normal University, Xi’an 710119, China; zhangcy@snnu.edu.cn (C.Z.); guozhihua@snnu.edu.cn (Z.G.)

**Keywords:** quantum coherence, measure, lower bound, upper bound

## Abstract

Quantum coherence is an important physical resource in quantum information science, and also as one of the most fundamental and striking features in quantum physics. To quantify coherence, two proper measures were introduced in the literature, the one is the relative entropy of coherence $C_{r}\left(\rho \right)=S\left(\rho_{\mathrm{diag}}\right)-S\left(\rho \right)$ and the other is the $\ell_{1}$-norm of coherence $C_{\ell_{1}}\left(\rho \right)=\sum_{i\neq j}\left|\rho_{ij}\right|$. In this paper, we obtain a symmetry-like relation of relative entropy measure $C_{r}\left(\rho^{A_{1}A_{2}\cdots A_{n}}\right)$ of coherence for an *n*-partite quantum states $\rho^{A_{1}A_{2}\cdots A_{n}}$, which gives lower and upper bounds for $C_{r}\left(\rho \right)$. As application of our inequalities, we conclude that when each reduced states $\rho^{A_{i}}$ is pure, $\rho^{A_{1}\cdots A_{n}}$ is incoherent if and only if the reduced states $\rho^{A_{i}}$ and ${\mathrm{tr}}_{A_{i}}\rho^{A_{1}\cdots A_{n}}\left(i=1,2,\ldots ,n\right)$ are all incoherent. Meanwhile, we discuss the conjecture that $C_{r}\left(\rho \right)\leq C_{\ell_{1}}\left(\rho \right)$ for any state ρ, which was proved to be valid for any mixed qubit state and any pure state, and open for a general state. We observe that every mixture η of a state ρ satisfying the conjecture with any incoherent state σ also satisfies the conjecture. We also observe that when the von Neumann entropy is defined by the natural logarithm ln instead of $\log_{2}$, the reduced relative entropy measure of coherence ${\bar{C}}_{r}\left(\rho \right)=-\rho_{\mathrm{diag}}\ln\rho_{\mathrm{diag}}+\rho \ln\rho$ satisfies the inequality ${\bar{C}}_{r}\left(\rho \right)\leq C_{\ell_{1}}\left(\rho \right)$ for any state ρ.

## 1. Introduction

Quantum computing utilizes the superposition and entanglement of quantum states to operate and process information. Its most significant advantage lies in the parallelism of operations [1,2,3]. To achieve efficient parallel computing in quantum computers, quantum coherence is essentially used. Quantum coherence arising from quantum superposition plays a central role in quantum mechanics and so becomes an important physical resource in quantum information and quantum computation [4]. It also plays an important role in a wide variety of research fields, such as quantum biology [5,6,7,8,9,10], nanoscale physics [11,12], and quantum metrology [13,14].

In 2014, Baumgratz et al. [15] proposed a framework to quantify coherence. In their seminal work, conditions that a suitable measure of coherence should satisfy have been put forward, including nonnegativity, the monotonicity under incoherent completely positive and trace preserving operations, the monotonicity under selective incoherent operations on average and the convexity under mixing of states. By introducing such a rigorous theoretical framework, a mass of properties and operations of quantification of coherence were discussed. Moreover, based on that framework, many coherence measures have been found, such as $\ell_{1}$-norm of coherence and relative entropy of coherence [15], fidelity and trace norm distances for quantifying coherence [16], robustness of coherence [17], geometric measure of coherence [18], coherence of formation [19], relative quantum coherence [20], measuring coherence with entanglement concurrence [21], trace distance measure of coherence [22,23,24].

In addition, some research related to quantum coherence have been developed, including quantum coherence and quantum correlations [25,26,27,28,29,30], an uncertainly-like relation about coherence [31], distribution of quantum coherence in multipartite systems [32], quantum coherence over the noisy quantum channels [33], maximally coherent mixed states [34], ordering states with coherence measures [35], coherence and path information [36], complementarity relations for quantum coherence [37], converting coherence to quantum correlations [38] and logarithmic coherence [39], quantum coherence and geometric quantum discord [40]. Recently, Guo and Cao [41] discussed the question of creating quantum correlation from a coherent state via incoherent quantum operations and obtained explicit interrelations among incoherent operations (IOs), maximally incoherent operations, genuinely incoherent operations and coherence breaking operations.

In this paper, we discuss some inequalities on the measures of quantum coherence. The organization of this paper is as follows: In Section 2, we recall the framework of coherence measure and basic properties of quantum coherence. In Section 3, we establish lower and upper bounds for the relative entropy measure of coherence in a multipartite system. In Section 4, we discuss the relation between $C_{r}\left(\rho \right)$ and $C_{\ell_{1}}\left(\rho \right)$. In Section 5, we give our conclusions obtained in this paper.

## 2. Preliminaries

In this section, we give a review of some fundamental notions about quantification of coherence, such as incoherence states, incoherence operations, and measures of coherence.

Let *H* be a *d*-dimensional Hilbert space, whose elements are denoted by the Dirac notations |ψ〉,|x〉 and so on, and let B(H) be the $C^{*}$-algebra consisting of all bounded linear operators on *H*. The adjoint operator of an operator *T* in B(H) is denoted by $T^{†}$. The identity operator on *H* is denoted by $I_{H}$, or simply, *I*. We use D(H) to denote the set of all density operators (positive and trace-1 operators) on *H*, whose elements are said to be the states of the quantum system *S* described by *H*. Fixed an orthonormal basis (ONB) $e=\{|e_{i}\rangle {\}}_{i=1}^{d}$ for *H*, a state ρ of *S* is said to be *incoherent* with respect to (w.r.t.) the basis *e* if $\langle e_{i}|\rho |e_{j}\rangle =0\left(i\neq j\right)$. Otherwise, it is said to be *coherent* w.r.t. *e*. Let I(e) be the set of all states of *S* that are incoherent w.r.t. *e*, that is,
$$I\left(e\right)=\left\{\rho \in D\left(H\right):\langle e_{i}|\rho |e_{j}\rangle =0\left(i\neq j\right)\right\}.$$

For every ρ∈D(H), we define
$$\rho_{e-\mathrm{diag}}=\sum_{i=1}^{d}\langle e_{i}\left|\rho \right|e_{i}\rangle |e_{i}\rangle \langle e_{i}|.$$
Clearly, $\rho_{e-\mathrm{diag}}\in I\left(e\right).$ By definition, a state ρ is incoherent w.r.t *e* if and only if $\rho =\rho_{e-\mathrm{diag}}$, i.e., it has a diagonal matrix representation w.r.t. *e*, i.e.,
$$\rho =\sum_{i=1}^{d}\lambda_{i}|e_{i}\rangle \langle e_{i}|\equiv \left(\begin{array}{cccc} \lambda_{1} & & & \\ & \lambda_{2} & & \\ & & \ddots & \\ & & & \lambda_{d} \end{array}\right),$$
where $\lambda_{i}\geq 0$ are eigenvalues of ρ and $\sum_{i=1}^{d}\lambda_{i}=\mathrm{tr}\rho =1;$ it is coherent w.r.t. *e* if and only if it can not be written as a diagonal matrix under this basis.

According to [42], a linear map E on the $C^{*}$-algebra B(H) is a completely positive and trace preserving (CPTP) map if and only if there exists a set of operators $K_{1},\ldots ,K_{m}$ in B(H) (called Kraus operators of E) with $\sum_{n=1}^{m}K_{n}^{†}K_{n}=I_{H}$ such that
$$E\left(T\right)=\sum_{n=1}^{m}K_{n}T{K_{n}}^{†},\;\;\forall T\in B\left(H\right).$$

A CPTP map E on B(H) is said to be an *e*-*incoherent operation (IO)* if it has Kraus operators $K_{1},\ldots ,K_{m}$ such that for all n=1,2,…,m, it holds that
$$K_{n}\rho {K_{n}}^{†}\in \mathrm{tr}\left(K_{n}\rho {K_{n}}^{†}\right)I\left(e\right),\;\;\forall \rho \in I\left(e\right).$$

In this case, we call ${\left\{K_{n}\right\}}_{n=1}^{m}$ a set of *e*-incoherent Kraus operators of E.

In order to measure coherence, Baumgratz et al. [15] presented the following four defining conditions for a coherence measure $C^{e}$:

$\left(A_{1}\right)\;C^{e}\left(\rho \right)\geq 0,\;\forall \rho \in D\left(H\right);\;\mathrm{and}\;C^{e}\left(\rho \right)=0$ if and only if ρ∈I(e).

$\left(A_{2}\right)\;C^{e}\left(\rho \right)\geq \;C^{e}\left(E\left(\rho \right)\right)$ for any *e*-incoherent operation E and any state ρ∈D(H).

$\left(A_{3}\right)\;C^{e}\left(\rho \right)\geq \sum_{n}p_{n}C^{e}\left(\rho_{n}\right)$ for any *e*-incoherent operation E with a set of *e*-incoherent Kraus operators $\left\{K_{n}\right\}$ and any state ρ∈D(H) where $\rho_{n}={p_{n}}^{-1}K_{n}\rho K_{n}^{†}$ with $p_{n}=\mathrm{tr}\left(K_{n}\rho K_{n}^{†}\right)\neq 0.$

$\left(A_{4}\right)\;\sum_{i}p_{i}C^{e}\left(\rho_{i}\right)\geq C^{e}\left(\sum_{i}p_{i}\rho_{i}\right)$ for any ensemble $\{p_{i},\rho_{i}\}.$

It was proved in [15] that the relative entropy $C_{r}^{e}\left(\rho \right)$ and the $\ell_{1}$-norm measure $C_{\ell_{1}}^{e}\left(\rho \right)$ of coherence satisfy these defining conditions, which are defined as follows:(1)$$C_{r}^{e}\left(\rho \right)=S\left(\rho_{e-\mathrm{diag}}\right)-S\left(\rho \right),$$
where S(ρ)=−tr(ρlogρ) is the von Neumann entropy, and
(2)$$C_{\ell_{1}}^{e}\left(\rho \right)=\sum_{i\neq j}|\langle e_{i}\left|\rho \right|e_{j}\rangle |.$$

Notably, for a bipartite quantum system AB, the reference basis for $H_{AB}=H_{A}⊗H_{B}$ can be taken as a local basis:$$e_{AB}:=e_{A}⊗e_{B}=\{|e_{i}\rangle ⊗|f_{k}\rangle |i=1,2,\ldots ,d_{A},k=1,2,\ldots ,d_{B}\},$$
where $e_{A}=\{|e_{i}\rangle {\}}_{i=1}^{d_{A}}$ and $e_{B}=\{|f_{k}\rangle {\}}_{k=1}^{d_{B}}$ are the orthonormal bases for $H_{A}$ and $H_{B}$, respectively. In this case, every $\rho^{AB}$ of AB has the following representation:(3)$$\rho^{AB}=\sum_{i,j=1}^{d_{A}}\sum_{k,l=1}^{d_{B}}\rho_{i,j,k,l}|e_{i}\rangle \langle e_{j}\left|⊗\right|f_{k}\rangle \langle f_{l}|.$$
Put
(4)$$\rho_{e_{AB}-\mathrm{diag}}^{AB}=\sum_{i=1}^{d_{A}}\sum_{k=1}^{d_{B}}\rho_{i,i,k,k}|e_{i}\rangle \langle e_{i}\left|⊗\right|f_{k}\rangle \langle f_{k}|.$$
Thus, a state $\rho^{AB}$ of the system AB is incoherent w.r.t. $e_{AB}$ if and only if $\rho^{AB}=\rho_{e_{AB}-\mathrm{diag}}^{AB}$, i.e.,
$$\rho_{i,j,k,l}:=\langle e_{i}|f_{k}|\rho^{AB}\left|e_{j}\right|f_{l}\rangle =0\left(\left(i,k\right)\neq \left(j,l\right)\right).$$

Moreover, let $\rho^{A}:={\mathrm{tr}}_{B}\left(\rho^{AB}\right)$ and $\rho^{B}:={\mathrm{tr}}_{A}\left(\rho^{AB}\right)$. Then from Equations (Equation 3) and (Equation 4), we get that
(5)$$\rho_{e_{A}-\mathrm{diag}}^{A}={\mathrm{tr}}_{B}\left(\rho_{e_{AB}-\mathrm{diag}}^{AB}\right),\rho_{e_{B}-\mathrm{diag}}^{B}={\mathrm{tr}}_{A}\left(\rho_{e_{AB}-\mathrm{diag}}^{AB}\right).$$

In next section, we derive some inequalities, which give lower and upper bounds for the relative entropy of coherence of multi-partite states.

## 3. Lower and Upper Bounds for the Relative Entropy of Coherence

Xi et al. [30] proved that for any bipartite quantum state $\rho^{AB}$, the relative entropy of coherence obeys some uncertainty-like relation by using the properties of relative entropy, which reads
(6)$$C_{r}^{e_{AB}}\left(\rho^{AB}\right)\geq C_{r}^{e_{A}}\left(\rho^{A}\right)+C_{r}^{e_{B}}\left(\rho^{B}\right),$$
where $\rho^{A}={\mathrm{tr}}_{B}\rho^{AB}$, $\rho^{B}={\mathrm{tr}}_{A}\rho^{AB}.$

Afterwards, Liu et al. [31] proved that any tripartite pure state $\rho^{ABC}$ satisfies
(7)$$C_{r}^{e_{ABC}}\left(\rho^{ABC}\right)\geq C_{r}^{e_{AB}}\left(\rho^{AB}\right)+C_{r}^{e_{AC}}\left(\rho^{AC}\right),$$
where $e_{ABC}:=e_{A}⊗e_{B}⊗e_{C},e_{AB}:=e_{A}⊗e_{B},e_{AC}:=e_{A}⊗e_{C},$
$\rho^{AB}={\mathrm{tr}}_{C}\rho^{ABC}$ and $\rho^{AC}={\mathrm{tr}}_{B}\rho^{ABC}$, provided that
(8)$$\lambda S\left(\rho_{e-\mathrm{diag}}^{AB}\right)\leq S\left(\rho^{AB}\right),\left(1-\lambda \right)S\left(\rho_{e-\mathrm{diag}}^{AC}\right)\leq S\left(\rho^{AC}\right)$$
for some 0≤λ≤1. Combining Equations (Equation 6) and (Equation 7), the following inequality was derived in [31]:(9)$$C_{r}^{e_{ABC}}\left(\rho^{ABC}\right)\geq \frac{4}{3}\left(C_{r}^{e_{A}}\left(\rho^{A}\right)+C_{r}^{e_{B}}\left(\rho^{B}\right)+C_{r}^{e_{C}}\left(\rho^{C}\right)\right)$$
for a pure state $\rho^{ABC}$ satisfying the condition (Equation 8).

The aim of this section is to establish lower and upper bounds of $C_{r}\left(\rho^{A_{1}A_{2}\cdots A_{n}}\right)$ for a general *n*-partite state $\rho^{A_{1}A_{2}\cdots A_{n}}$. To do this, we use $\rho_{\mathrm{diag}}^{X}$ and $C_{r}\left(\rho^{X}\right)$ to denote $\rho_{e_{X}-\mathrm{diag}}^{X}$ and $C_{r}^{e_{X}}\left(\rho^{X}\right),$ respectively.

First, for a bipartite $\rho^{AB}$ of the system AB, we know from Equation (Equation 5) and the subadditivity of von Neumann entropy that
$$S\left(\rho_{\mathrm{diag}}^{AB}\right)\leq S\left({\mathrm{tr}}_{B}\rho_{\mathrm{diag}}^{AB}\right)+S\left({\mathrm{tr}}_{A}\rho_{\mathrm{diag}}^{AB}\right)=S\left(\rho_{\mathrm{diag}}^{A}\right)+S\left(\rho_{\mathrm{diag}}^{B}\right)$$
and so
$$\begin{array}{ll} & C_{r}\left(\rho^{AB}\right)-C_{r}\left(\rho^{A}\right)-C_{r}\left(\rho^{B}\right)\\ = & S\left(\rho_{\mathrm{diag}}^{AB}\right)-S\left(\rho^{AB}\right)-S\left(\rho_{\mathrm{diag}}^{A}\right)+S\left(\rho^{A}\right)-S\left(\rho_{\mathrm{diag}}^{B}\right)+S\left(\rho^{B}\right)\\ \leq & S\left(\rho_{\mathrm{diag}}^{A}\right)+S\left(\rho_{\mathrm{diag}}^{B}\right)-S\left(\rho^{AB}\right)-S\left(\rho_{\mathrm{diag}}^{A}\right)+S\left(\rho^{A}\right)-S\left(\rho_{\mathrm{diag}}^{B}\right)+S\left(\rho^{B}\right)\\ = & S\left(\rho^{A}\right)+S\left(\rho^{B}\right)-S\left(\rho^{AB}\right). \end{array}$$
Thus,
$$C_{r}\left(\rho^{AB}\right)\leq C_{r}\left(\rho^{A}\right)+C_{r}\left(\rho^{B}\right)+S\left(\rho^{A}\right)+S\left(\rho^{B}\right)-S(\rho^{AB}).$$
Combing this with Equation (Equation 6), we have
(10)$$\frac{1}{2}\left(2C_{r}\left(\rho^{A}\right)+2C_{r}\left(\rho^{B}\right)\right)\leq C_{r}\left(\rho^{AB}\right)\leq C_{r}\left(\rho^{A}\right)+C_{r}\left(\rho^{B}\right)+S\left(\rho^{A}\right)+S\left(\rho^{B}\right)-S(\rho^{AB}).$$

Second, for a tripartite quantum state $\rho^{ABC}$, according to the super-additivity inequality (Equation 6), we have
$$C_{r}\left(\rho^{ABC}\right)\geq C_{r}\left(\rho^{A}\right)+C_{r}\left(\rho^{BC}\right),$$
$$C_{r}\left(\rho^{ABC}\right)\geq C_{r}\left(\rho^{B}\right)+C_{r}\left(\rho^{AC}\right),$$
$$C_{r}\left(\rho^{ABC}\right)\geq C_{r}\left(\rho^{C}\right)+C_{r}\left(\rho^{AB}\right).$$
By finding the sums of two sides of the inequalities above, we obtain
(11)$$C_{r}\left(\rho^{ABC}\right)\geq \frac{1}{3}\left(C_{r}\left(\rho^{AB}\right)+C_{r}\left(\rho^{BC}\right)+C_{r}\left(\rho^{AC}\right)+C_{r}\left(\rho^{A}\right)+C_{r}\left(\rho^{B}\right)+C_{r}\left(\rho^{C}\right)\right).$$
On the other hand, using definition (1) yields that
$$\begin{array}{ll} & C_{r}\left(\rho^{ABC}\right)-C_{r}\left(\rho^{AB}\right)-C_{r}\left(\rho^{AC}\right)-C_{r}\left(\rho^{BC}\right)\\ = & S\left(\rho_{\mathrm{diag}}^{ABC}\right)-S\left(\rho^{ABC}\right)-S\left(\rho_{\mathrm{diag}}^{AB}\right)+S\left(\rho^{AB}\right)\\ & -S\left(\rho_{\mathrm{diag}}^{BC}\right)+S\left(\rho^{BC}\right)-S\left(\rho_{\mathrm{diag}}^{AC}\right)+S\left(\rho^{AC}\right)\\ = & \left[S\left(\rho_{\mathrm{diag}}^{ABC}\right)+S\left(\rho_{\mathrm{diag}}^{B}\right)-S\left(\rho_{\mathrm{diag}}^{AB}\right)-S\left(\rho_{\mathrm{diag}}^{BC}\right)\right]\\ & +\left[S\left(\rho^{AC}\right)-S\left(\rho_{\mathrm{diag}}^{AC}\right)\right]+\left[S\left(\rho^{AB}\right)+S\left(\rho^{BC}\right)-S\left(\rho^{ABC}\right)-S\left(\rho_{\mathrm{diag}}^{B}\right)\right]\\ \leq & S\left(\rho^{A}\right)+S\left(\rho^{B}\right)+S\left(\rho^{B}\right)+S\left(\rho^{C}\right)-S\left(\rho_{\mathrm{diag}}^{B}\right)-S\left(\rho^{ABC}\right)\\ \leq & S\left(\rho^{A}\right)+S\left(\rho^{B}\right)+S\left(\rho^{C}\right)-S(\rho^{ABC}), \end{array}$$
since $S\left(\rho_{\mathrm{diag}}^{ABC}\right)+S\left(\rho_{\mathrm{diag}}^{B}\right)-S\left(\rho_{\mathrm{diag}}^{AB}\right)-S\left(\rho_{\mathrm{diag}}^{BC}\right)\leq 0$ (strong subadditivity) and $S\left(\rho^{AC}\right)-S\left(\rho_{\mathrm{diag}}^{AC}\right)\leq 0$. This shows that
(12)$$C_{r}\left(\rho^{ABC}\right)\leq C_{r}\left(\rho^{AB}\right)+C_{r}\left(\rho^{AC}\right)+C_{r}\left(\rho^{BC}\right)+S\left(\rho^{A}\right)+S\left(\rho^{B}\right)+S\left(\rho^{C}\right)-S(\rho^{ABC}).$$
Combining Equations (Equation 11) and (Equation 12) gives
(13)$$\begin{array}{l} \frac{1}{3}\left(C_{r}\left(\rho^{AB}\right)+C_{r}\left(\rho^{AC}\right)+C_{r}\left(\rho^{BC}\right)+C_{r}\left(\rho^{A}\right)+C_{r}\left(\rho^{B}\right)+C_{r}\left(\rho^{C}\right)\right)\\ \leq C_{r}\left(\rho^{ABC}\right)\\ \leq C_{r}\left(\rho^{AB}\right)+C_{r}\left(\rho^{AC}\right)+C_{r}\left(\rho^{BC}\right)+S\left(\rho^{A}\right)+S\left(\rho^{B}\right)+S\left(\rho^{C}\right)-S(\rho^{ABC}). \end{array}$$

As a generalization of inequalities (Equation 10) and (Equation 13), we can prove the following inequalities (Equation 14) for any *n*-partite state $\rho^{A_{1}\cdots A_{n}}$ of the system $H_{A_{1}A_{2}\cdots A_{n}}=H_{A_{1}}⊗H_{A_{2}}⊗\cdots ⊗H_{A_{n}}$, which give lower and upper bounds for the relative entropy of coherence. To do this, we let $e_{A_{k}}=\{|e_{i_{k}}^{k}\rangle {\}}_{i_{k}=1}^{d_{k}}$ be an orthogonal basis for the Hilbert space $H_{A_{k}}\left(k=1,2,\ldots ,n\right)$, and let
$$e_{A_{1}A_{2}\cdots A_{n}}=\{|e_{i_{1}}^{1}\rangle |e_{i_{2}}^{2}\rangle \cdots |e_{i_{n}}^{n}\rangle :\;1\leq i_{k}\leq d_{k}\left(k=1,2,\ldots ,n\right)\},$$
which is an orthogonal basis for the Hilbert space $H_{A_{1}A_{2}\cdots A_{n}}$. Thus,
$$e_{A_{2}\cdots A_{n}}=\{|e_{i_{2}}^{2}\rangle |e_{i_{3}}^{3}\rangle \cdots |e_{i_{n}}^{n}\rangle :\;1\leq i_{k}\leq d_{k}\left(k=2,3,\ldots ,n\right)\}$$
becomes an orthogonal basis for the Hilbert space $H_{A_{2}A_{3}\cdots A_{n}}=H_{A_{2}}⊗H_{A_{3}}⊗\cdots ⊗H_{A_{n}}$. With these notations, we have the following.

**Theorem** **1.**
*For any state $\rho^{A_{1}\cdots A_{n}}$ of the system $H_{A_{1}A_{2}\cdots A_{n}}=H_{A_{1}}⊗H_{A_{2}}⊗\cdots ⊗H_{A_{n}}$, it holds that*
(14)$$\frac{1}{n}\sum_{i=1}^{n}\left[C_{r}\left({\mathrm{tr}}_{A_{i}}\rho^{A_{1}\cdots A_{n}}\right)+C_{r}\left(\rho^{A_{i}}\right)\right]\leq C_{r}\left(\rho^{A_{1}\cdots A_{n}}\right)\leq \sum_{i=1}^{n}\left[C_{r}\left({\mathrm{tr}}_{A_{i}}\rho^{A_{1}\cdots A_{n}}\right)+S\left(\rho^{A_{i}}\right)\right]-S(\rho^{A_{1}\cdots A_{n}}),$$
*where $\rho^{A_{i}}$ denotes the reduced state of $\rho^{A_{1}\cdots A_{n}}$ on the subsystem $A_{i}$.*


**Proof.** To prove that the first inequality in Equation (Equation 14) holds, we know from Equation (Equation 6) that$$\begin{array}{c} C_{r}\left(\rho^{A_{1}\cdots A_{n}}\right)\geq C_{r}\left(\rho^{A_{1}}\right)+C_{r}\left({\mathrm{tr}}_{A_{1}}\rho^{A_{1}\cdots A_{n}}\right),\\ C_{r}\left(\rho^{A_{1}\cdots A_{n}}\right)\geq C_{r}\left(\rho^{A_{2}}\right)+C_{r}\left({\mathrm{tr}}_{A_{2}}\rho^{A_{1}\cdots A_{n}}\right),\\ \vdots \\ C_{r}\left(\rho^{A_{1}\cdots A_{n}}\right)\geq C_{r}\left(\rho^{A_{n}}\right)+C_{r}\left({\mathrm{tr}}_{A_{n}}\rho^{A_{1}\cdots A_{n}}\right), \end{array}$$
and consequently,
$$C_{r}\left(\rho^{A_{1}\cdots A_{n}}\right)\geq \frac{1}{n}\left(\sum_{i=1}^{n}C_{r}\left({\mathrm{tr}}_{A_{i}}\rho^{A_{1}\cdots A_{n}}\right)+\sum_{i=1}^{n}C_{r}\left(\rho^{A_{i}}\right)\right).$$Next, let us prove that the second inequality in (Equation 14) holds by using mathematical induction. Firstly, we know from Equation (Equation 10) that the desired inequality holds for n=2 and any bipartite state. Secondly, we assume the second inequality in (Equation 14) holds for n=N−1 and any N−1-partite state. Then for any *N*-partite state $\rho^{A_{1}\cdots A_{N}}$, we have
$$\begin{array}{ccc} C_{r}\left(\rho^{A_{1}\cdots A_{N}}\right) & = & C_{r}\left(\rho^{A_{1}\cdots A_{N-2}\left(A_{N-1}A_{N}\right)}\right)\\ & \leq & \sum_{i=1}^{N-2}C_{r}\left({\mathrm{tr}}_{A_{i}}\rho^{A_{1}\cdots A_{N-2}\left(A_{N-1}A_{N}\right)}\right)+C_{r}\left({\mathrm{tr}}_{A_{N-1}A_{N}}\rho^{A_{1}\cdots A_{N}}\right)\\ & & +\sum_{i=1}^{N-2}S\left(\rho^{A_{i}}\right)+S\left(\rho^{A_{N-1}A_{N}}\right)-S(\rho^{A_{1}\cdots A_{N-2}\left(A_{N-1}A_{N}\right)}). \end{array}$$
By using Equation (Equation 6), we know that $C_{r}\left({\mathrm{tr}}_{X}\eta \right)\leq C_{r}\left(\eta \right)$. Thus,
$$C_{r}\left({\mathrm{tr}}_{A_{N-1}A_{N}}\rho^{A_{1}\cdots A_{N}}\right)\leq C_{r}\left({\mathrm{tr}}_{A_{N}}\rho^{A_{1}\cdots A_{N}}\right)\leq C_{r}\left({\mathrm{tr}}_{A_{N-1}}\rho^{A_{1}\cdots A_{N}}\right)+C_{r}\left({\mathrm{tr}}_{A_{N}}\rho^{A_{1}\cdots A_{N}}\right).$$
Combining the fact that
$$S\left(\rho^{A_{N-1}A_{N}}\right)\leq S\left(\rho^{A_{N-1}}\right)+S\left(\rho^{A_{N}}\right),S\left(\rho^{A_{1}\cdots A_{N-2}\left(A_{N-1}A_{N}\right)}\right)=S\left(\rho^{A_{1}\cdots A_{N}}\right),$$
we get that
$$\begin{array}{ccc} C_{r}\left(\rho^{A_{1}\cdots A_{N}}\right) & \leq & \sum_{i=1}^{N-2}C_{r}\left({\mathrm{tr}}_{A_{i}}\rho^{A_{1}\cdots A_{N-2}\left(A_{N-1}A_{N}\right)}\right)+C_{r}\left({\mathrm{tr}}_{A_{N-1}}\rho^{A_{1}\cdots A_{N}}\right)\\ & & +C_{r}\left({\mathrm{tr}}_{A_{N}}\rho^{A_{1}\cdots A_{N}}\right)+\sum_{i=1}^{N-2}S\left(\rho^{A_{i}}\right)+S\left(\rho^{A_{N-1}}\right)+S\left(\rho^{A_{N}}\right)-S(\rho^{A_{1}\cdots A_{N}})\\ & = & \sum_{i=1}^{N}C_{r}\left({\mathrm{tr}}_{A_{i}}\rho^{A_{1}\cdots A_{N}}\right)+\sum_{i=1}^{N}S\left(\rho^{A_{i}}\right)-S(\rho^{A_{1}\cdots A_{N}}). \end{array}$$
Thus, the validity of the second inequality in Equation (Equation 14) is proved. The proof is completed. □

As immediate application of Theorem 1, we have the following corollaries.

**Corollary** **1.**
*Let $\rho^{A_{1}\cdots A_{n}}$ be a state of the system $H_{A_{1}A_{2}\cdots A_{n}}=H_{A_{1}}⊗H_{A_{2}}⊗\cdots ⊗H_{A_{n}}$. If $\rho^{A_{1}\cdots A_{n}}$ is incoherent, then the reduced states $\rho^{A_{i}}$ and ${\mathrm{tr}}_{A_{i}}\rho^{A_{1}\cdots A_{n}}\left(i=1,2,\ldots ,n\right)$ are all incoherent. The converse is true if each reduced states $\rho^{A_{i}}$ is pure.*


**Corollary** **2.**
*Let $\rho^{A_{1}\cdots A_{n}}$ be a state of the system $H_{A_{1}A_{2}\cdots A_{n}}=H_{A_{1}}⊗H_{A_{2}}⊗\cdots ⊗H_{A_{n}}$ such that the reduced states $\rho^{A_{i}}\left(i=1,2,\ldots ,n\right)$ are pure and incoherent. Then*
(15)$$\frac{1}{n}\sum_{i=1}^{n}C_{r}\left({\mathrm{tr}}_{A_{i}}\rho^{A_{1}\cdots A_{n}}\right)\leq C_{r}\left(\rho^{A_{1}\cdots A_{n}}\right)\leq \sum_{i=1}^{n}C_{r}\left({\mathrm{tr}}_{A_{i}}\rho^{A_{1}\cdots A_{n}}\right).$$


It is remarkable that the equalities in Equation (Equation 14) may hold in some cases. For example, when $d_{1}=d_{2}=\cdots =d_{n}=d$ and
(16)$$|\psi_{A_{1}\cdots A_{n}}\rangle ={\left(\frac{1}{\sqrt{d}}\right)}^{n}\sum_{i_{1},i_{2},\cdots ,i_{n}=1}^{d}|e_{i_{1}}^{1}\rangle |e_{i_{2}}^{2}\rangle \cdots |e_{i_{n}}^{n}\rangle ,$$
the maximally coherent state
$$\rho^{A_{1}A_{2}\cdots A_{n}}=|\psi_{A_{1}\cdots A_{n}}\rangle \langle \psi_{A_{1}\cdots A_{n}}|$$
satisfies
$$\frac{1}{n}\sum_{i=1}^{n}\left[C_{r}\left({\mathrm{tr}}_{A_{i}}\rho^{A_{1}\cdots A_{n}}\right)+C_{r}\left(\rho^{A_{i}}\right)\right]=C_{r}\left(\rho^{A_{1}\cdots A_{n}}\right)=n\log_{2}d,$$
due to the fact that $C_{r}\left(\rho^{A_{j}}\right)=\log_{2}d$ for j=1,2,…,n, and
$$C_{r}\left(\rho^{A_{1}\cdots A_{n}}\right)=-\frac{1}{d^{n}}\log_{2}\frac{1}{d^{n}}\times d^{n}=n\log_{2}d,\;C_{r}\left({\mathrm{tr}}_{A_{i}}\rho^{A_{1}\cdots A_{n}}\right)=\left(n-1\right)\log_{2}d.$$
Moreover, the second inequality in Equation (Equation 14) also becomes equality when n=2. This shows that the inequalities in Equation (Equation 14) are tight and can not be improved.

## 4. The Relation between $C_{r}\left(\rho \right)$ and $C_{\ell_{1}}\left(\rho \right)$

In this section, we discuss the relation between $C_{r}\left(\rho \right)$ and $C_{\ell_{1}}\left(\rho \right)$. Rana et al. found that the inequality
(17)$$C_{r}\left(\rho \right)\leq C_{\ell_{1}}\left(\rho \right)$$
holds for any mixed qubit state ([39], Proposition 1) and any pure state ([39], Proposition 3). Moreover, they conjectured that the inequality (Equation 17) holds for all states ρ. It was also proved ([39], Proposition 6) that inequality (Equation 17) holds for any state ρ of the form ρ=p|ψ〉〈ψ|+(1−p)δ(0≤p≤1) provided that δ is an incoherent state w.r.t. the reference basis. As an extension of this result, we have the following.

**Proposition** **1.**
*Let ρ be a state of S satisfying Equation (Equation 17) and let σ be any incoherent state of S. Then every mixture η:=pρ+(1−p)σ(0≤p≤1) of ρ and σ satisfies (17).*


**Proof.** The convexity of $C_{r}$ implies that
$$\begin{array}{ccc} C_{r}\left(\eta \right) & \leq & pC_{r}\left(\rho \right)+\left(1-p\right)C_{r}\left(\sigma \right)\\ & = & pC_{r}\left(\rho \right)\\ & \leq & pC_{\ell_{1}}\left(\rho \right)\\ & = & C_{\ell_{1}}\left(p\rho +\left(1-p\right)\sigma \right)\\ & = & C_{\ell_{1}}\left(\eta \right). \end{array}$$
The proof is completed. □

Rana et al. proved in ([22], Proposition 6) that for arbitrary state ρ of a *d*-dimensional system, it holds that
(18)$$C_{r}\left(\rho \right)\leq C_{\ell_{1}}\left(\rho \right)\log_{2}d$$
and derived in ([39], Equation (Equation 10)) that
(19)$$\begin{array}{c} C_{r}\left(\rho \right)\leq \left\{\begin{array}{cc} C_{\ell_{1}}\left(\rho \right), & \;\mathrm{if}\;C_{\ell_{1}}\left(\rho \right)\geq 1;\\ C_{\ell_{1}}\left(\rho \right)\log_{2}e, & \;\mathrm{if}\;C_{\ell_{1}}\left(\rho \right)<1. \end{array}\right. \end{array}$$
So, $C_{r}\left(\rho \right)\leq C_{\ell_{1}}\left(\rho \right)\log_{2}e$ for all ρ. Thus, if we redefine the von Neumann entropy as $\bar{S}\left(\rho \right)=-\mathrm{tr}\left(\rho \ln\rho \right)$, then the resulted relative entropy of coherence reads
(20)$${\bar{C}}_{r}\left(\rho \right)=\bar{S}\left(\rho_{\mathrm{diag}}\right)-\bar{S}\left(\rho \right)=\frac{1}{\log_{2}e}C_{r}\left(\rho \right).$$
This leads to the following inequality:(21)$${\bar{C}}_{r}\left(\rho \right)\leq C_{\ell_{1}}\left(\rho \right),\;\;\forall \rho \in D\left(H\right).$$

## 5. Conclusions

In this paper, we have established lower and upper bounds for relative entropy of coherence $C_{r}\left(\rho^{A_{1}A_{2}\cdots A_{n}}\right)$ for an *n*-partite quantum states $\rho^{A_{1}A_{2}\cdots A_{n}}$. As application of our inequalities, we have found that when each reduced states $\rho^{A_{i}}$ is pure, $\rho^{A_{1}\cdots A_{n}}$ is incoherent if and only if the reduced states $\rho^{A_{i}}$ and ${\mathrm{tr}}_{A_{i}}\rho^{A_{1}\cdots A_{n}}\left(i=1,2,\ldots ,n\right)$ are all incoherent. Moreover, we have discussed the conjecture that $C_{r}\left(\rho \right)\leq C_{\ell_{1}}\left(\rho \right)$ for any state ρ and observed that every mixture η of a state ρ satisfying the conjecture with any incoherent state σ also satisfies the conjecture. We have also proved that when the von Neumann entropy is defined by the natural logarithm ln instead of $\log_{2}$, the reduced relative entropy measure of coherence ${\bar{C}}_{r}\left(\rho \right)=-\rho_{\mathrm{diag}}\ln\rho_{\mathrm{diag}}+\rho \ln\rho$ satisfies the inequality ${\bar{C}}_{r}\left(\rho \right)\leq C_{\ell_{1}}\left(\rho \right)$ for any state ρ.

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
