# Peer review of "Symmetry-Like Relation of Relative Entropy Measure of Quantum Coherence"

_entropy, 2020, doi:10.3390/e22030297_

Round 1
Reviewer 1 Report
The authors have verified some relations about relative entropy coherence as well its connection to the l1-norm of coherence. I find the work to be consistent and good enough for publication as it is.
Author Response
Dear the reviewer,
Thank you for your kind comments and suggestions.
(1) English language and style have been checked and revised.
(2) According to another reviewer's comments, the following revision have been made.
- In the abstract, notations $C_{\ell_1}(\rho)$ and $C_r(\rho)$ are defined and an application of our theorem is pointed out.
- Some general notations are given in Section 2.
- Before Theorem 3.1, some motivations and notations are given.
- Theorem 3.1 is re-described and Inequalities (11) are rewritten as one sum in each side.
- Section 4 is rewritten and the original Example is removed.
- Three references [1-3] are added and cited in the Introduction part.
Thank you again.
H.X. Cao
Reviewer 2 Report
In this manuscript the authors have studied coherence measures---mainly the relative entropy of coherence (C_r), its bound for multipartite states and a conjectured relation between C_r and C_{\ell_1}. Although the manuscript has a long introduction spanning the first two sections (too long for my taste, but I understand the authors may want to keep everything self-contained in their article), my overall assessment of the manuscript is that it seriously lacks contents (enough and/or novel) to warrant publication. The results reported are very simple, all follow merely from definition of the quantities and there is almost nothing new. Crucially, most of the results have already been reported in the literature (I assume the authors did not pay enough attention to the literature, especially while citing). Let me give a detail explanation section wise.
The first two sections comprise of Introduction and Preliminaries, so there is nothing novel to comment on.The first result is the lower and upper bound on C_r for a multipartite state, in Theorem 3.1 of Section 3. First of all, there are typos: There should be only one summation on each of the bounds (instead of the current two different summations over i), otherwise it does not make any sense.
Now let’s see why both these bounds are trivial. Although these are for multipartite states, it is exactly derived directly from the bipartite state. For example, the upper bound on the left follows immediately from Eq (3) [which has been shown Ref. 27, btw nothing surprising, as it is a direct consequence of the sub-additivity of entropy function], just take A as one of the n subsystems and the rest of the system as B and sum over. Same goes for the other bound.
For the same reason, I think that both the bounds are far from optimal (sharp), and could be improved further, e.g., using strong sub additivity of entropy.
Now for the section 4, unfortunately there is nothing new here, everything has been reported in the literature that the authors have already cited.
The Proposition 4.1 is an extended version of Proposition 6 from the Ref. [36], it uses the same logic and proof. I do not see any need to consider the special example of maximally incoherent mixed states thereafter. As it is a mixture of a pure state and a diagonal state, it follows from Proposition 6 of the Ref. [36] that these states will follow the conjecture.
Thus, there is no new result in this section.
My final conclusion, based on my analysis above, is that the manuscript does not contain any novel idea or result to justify the scientific merit for publication. Hence I do not recommend publication of this manuscript.
Author Response
Dear the revewer,
Thank you for your kind comments and suggestions.
As you said, our main inequality (11) was obtained by using the Eq (3) established in Ref. [27](Now, [30]), but it was not found in the literature and so should be a new result although its proof is not difficulty. Indeed, every new result is always obtained based on some known ones.
As you said, our Proposition 4.1 is really an extended version of Proposition 6 from the Ref. [36](Now, [39] ), such an extension was not found in the literature and so should be a new result although its proof is similar to the original one.
According to another rivewer's comments, the following changes have been made.
- In the abstract, notations $C_{\ell_1}(\rho)$ and $C_r(\rho)$ are defined and an application of our theorem is pointed out.
- Some general notations are given in Section 2.
- Before Theorem 3.1, some motivations and notations are given.
- Theorem 3.1 is re-described and Inequalities (11) are rewritten as one sum in each side.
- Section 4 is rewritten and the original Example is removed.
- Three references [1-3] are added and cited in the Introduction part.
Thank you again.
H.X. Cao
Reviewer 3 Report
1. BRIEF SUMMARY
Since the formalization of the resource theory of quantum coherence, many other quantum properties have been identified as potential resources in quantum information science. But quantum coherence still represents one of the most important physical resources, and it has deserved a lot of investigation. The authors of this paper study two well-defined measures of quantum coherence, namely the relative entropy of coherence and the L1-norm measure. Specifically, they prove that for any multipartite state, the relative entropy of coherence of the joint state is lower and upper bounded by some quantities that combines the coherences of the marginal states and the entropies of the individual parties. Moreover, they prove two possible extensions of an order relation between the two measures, which was known to hold only for qubits or pure states.
2. BROAD COMMENTS
The article fits in the scope of Entropy, as the authors provide a couple of novel results regarding some information-theoretic measures of quantum coherence. Overall, I think the contribution is technically
correct and interesting, and I would recommend that this manuscript merits to be published in Entropy. There are some
amendments that must be fi xed before that, though.
I would suggest the authors to make some general changes in order to improve the readability and the impact. First, I some editing of language and style is needed. Second, about the content and its presentation, it should be checked:
- that some other references are needed,
- the order in which some notation is introduced (for example, the symbol for the L1-norm measure used in the abstract),
- the way in which the main results are presented.
I will give some detailed comments about all these points in the next section. Finally, it would be interesting to have some deeper explanation on the possible impact of this new results, maybe some simple application or example.
3. SPECIFIC COMMENTS
Before giving some minor comments, I would like to say something general about Section 3. In the statement of Theorem 3.1. you talk about arbitrary quantum states. However, before giving the enunciation, you go into details of several results that hold only for pure states. Maybe, it would be better if you emphasize that distinction in a clearer way.
- Line 5: The notation C_{\ell_1} has not been introduced yet.
- Line 5: It is not at all clear that the states \rho satisfying the inequality are qubits or pure. I think it is important say that somewhere in the abstract.
- Line 13: Any reference supporting the claim that “Its most significant advantage lies in the parallelism of operations”?
- Line 30: It should be “an uncertainty-like relation”
- Lines 39-45: Check the verbal tense. For example, you begin with “In this paper, we will discuss…”. After that, you say “In Section 2, we recall…”. And then, “In Section 4, we will…”.
- Lines 51-52: It should say “Let \mathcal{I}(e) be the set…”. Also, check the notation of what you call the “diagonal state of \rho” because you are using different dashes.
- Lines 55-56: Given that there are many different classes of incoherent operations in the literature, some brief remark about the one you are choosing should be desirable.
- Line 66: In the equation before this line, you are using a notation for the brackets that I am not sure to be standard.
- Line 67: It should say “In the next section”.
- Line 84: Before equation (13) it should say “and then proved the upper bound…”.
- Line 85: It should say “they conjectured that inequality…”.
- Line 86: In equation (14), you have some typos after the “if” conditions.
- Line 88: As in the abstract, I think that it would be better if you say that the state should be qubit or pure in the statement of Proposition 4.1.
Author Response
Dear the reviewer,
Thank you for your kind comments and suggestions.
To illustrate applications of our Theorem 3.1, two corollaries have been added followed the proof of the theorem. And a remark has been added in the end of Section 3 in order to shows that our inequalities are tight in some cases and then cannot be improved.
All of the points that you pointed out have been considered in the revision, see the following,
- In the abstract, notations $C_{\ell_1}(\rho)$ and $C_r(\rho)$ are defined and an application of our theorem is pointed out.
- Some general notations are given in Section 2.
- Before Theorem 3.1, some motivations and notations are given.
- Theorem 3.1 is re-described and Inequalities (11) are rewritten as one sum in each side.
- Section 4 is rewritten and the original Example is removed.
- Three references [1-3] are added and cited in the Introduction part.
Thank you again. Further comments and suggestions are welcome.
Best wishes,
H.X. Cao
Round 2
Reviewer 2 Report
As I previously reported, I do not find enough novel content to recommend publishing this manuscript. However, now looking at the response of my other two colleagues, I am totally confused. I must admit that I do not follow articles from entropy (nor going to publish anything there) so I have no idea about the authorship and readership of this journal.
I am happy with the response from the author(s), at least they realized that the example was unnecessary! In view of this, I would like to recommend publication of this manuscript in its present form.